# Optimizing Fluid Management Guided by Volumetric Parameters in Patients with Sepsis and ARDS

**DOI:** 10.3390/ijms24108768

**Published:** 2023-05-15

**Authors:** Evgeniia V. Fot, Natalia O. Khromacheva, Aleksei A. Ushakov, Aleksei A. Smetkin, Vsevolod V. Kuzkov, Mikhail Y. Kirov

**Affiliations:** Department of Anesthesiology and Intensive Care Medicine, Northern State Medical University, Arkhangelsk 163000, Russiaanesth_sm@mail.ru (A.A.S.); v_kuzkov@mail.ru (V.V.K.); mikhail_kirov@hotmail.com (M.Y.K.)

**Keywords:** de-escalation therapy, acute respiratory distress syndrome, sepsis, hemodynamics, extravascular lung water, global end-diastolic volume

## Abstract

We compared two de-escalation strategies guided by either extravascular lung water or global end-diastolic volume-oriented algorithms in patients with sepsis and ARDS. Sixty patients with sepsis and ARDS were randomized to receive de-escalation fluid therapy, guided either by the extravascular lung water index (EVLWI, *n* = 30) or the global end-diastolic volume index (GEDVI, *n* = 30). In cases of GEDVI > 650 mL/m^2^ or EVLWI > 10 mL/kg, diuretics and/or controlled ultrafiltration were administered to achieve the cumulative 48-h fluid balance in the range of 0 to −3000 mL. During 48 h of goal-directed de-escalation therapy, we observed a decrease in the SOFA score (*p* < 0.05). Extravascular lung water decreased only in the EVLWI-oriented group (*p <* 0.001). In parallel, PaO_2_/FiO_2_ increased by 30% in the EVLWI group and by 15% in the GEDVI group (*p <* 0.05). The patients with direct ARDS demonstrated better responses to dehydration therapy concerning arterial oxygenation and lung fluid balance. In sepsis-induced ARDS, both fluid management strategies, based either on GEDVI or EVLWI, improved arterial oxygenation and attenuated organ dysfunction. The de-escalation therapy was more efficient for direct ARDS.

## 1. Introduction

Optimization of fluid status is a fundamental challenge of critical care practice, particularly in patients with sepsis, shock, and acute respiratory distress syndrome (ARDS) [1]. Thus, only 50% of patients with septic shock demonstrate a clinically significant increase in stroke volume in response to a fluid bolus. Furthermore, those patients who were initially fluid responders rapidly become non-responsive to further escalation of the fluid load [2].

In many cases, liberal fluid therapy in the critically ill brings minimal hemodynamic benefits but can result in numerous side effects. There are two key mechanisms by which aggressive fluid therapy may be harmful to patients with sepsis. The first mechanism is related to the direct effects of excessive infusion on the cardiovascular system, worsening the shock. The second one includes the negative effects of volume overload on organ function [3]. Aggressive fluid therapy has been well-established to be a major risk factor for pulmonary edema, secondary intra-abdominal hypertension, acute kidney injury (AKI), hepatic dysfunction, multi-organ failure, and death [4,5,6,7,8]. The adverse effects of positive fluid balance become even more relevant during the systemic inflammation and capillary leak syndrome typical for sepsis and ARDS [9,10,11].

Taking into account these complications related to infusion therapy, the approach to fluid and vasopressor administration in cases of shock and ARDS should be individualized and based on the patient’s unique hemodynamic and clinical characteristics [10]. At the same time, there are still no universal goal-directed protocols for fluid therapy in patients with sepsis and ARDS [12]. However, it is established that maintenance of a timely negative fluid balance during the de-escalation phase of shock management can be associated with better outcomes [11]. Nevertheless, the heterogeneity of critically ill patients emphasizes the need for further investigations to determine the optimal fluid strategy. The results of these studies should address the appropriate timing for conservative fluid management, adequate hemodynamic goals and targets of de-escalation, and the benefits and harms of methods for dehydration, including diuretics and renal replacement therapy (RRT) [13].

We hypothesized that de-escalation fluid management using volumetric hemodynamic monitoring, based on parameters of interstitial and intravascular sectors, is safe and leads to the attenuation of organ dysfunction in patients with sepsis-induced ARDS. Thus, the aim of our study was to compare two de-escalation strategies guided by either extravascular lung water or global end-diastolic, volume-oriented algorithms in sepsis and ARDS.

## 2. Results

At baseline, both study groups were similar with respect to demographic characteristics, origin of sepsis and lung injury, severity of ARDS, organ dysfunction, use of vasopressors, diuretics and CVVH, and fluid balance (Table 1). At study enrollment, we observed the increased values of both GEDVI and EVLWI in 55% of patients. Five percent of patients had GEDVI < 650 mL/m^2^ and EVLWI < 10 mL/kg, and only in 40% of the patients, we found discordant changes in GEDVI and EVLWI (12% with EVLWI ≥ 10 mL/kg PBW and GEDVI< 650 mL/m^2^; 28% withGEDVI≥650 mL/m^2^ and EVLWI < 10 mL/kg PBW). During the 48 h of de-escalation therapy, we observed a decrease in the SOFA scores of the EVLWI-oriented (*p* = 0.038) and the GEDVI-oriented (*p* = 0.041) groups by 25% and 13%, respectively. The number of patients requiring continuous infusion of norepinephrine decreased by 53% during the EVLWI-guided de-escalation(*p* = 0.002) and by 43% during the GEDVI-guided de-escalation (*p* = 0.007). There were no statistically significant intergroup differences either in the duration of mechanical ventilation and ICU stay or in the time of hospitalization and the survival rate.

At the baseline, most hemodynamic and respiratory parameters (Table 2 and Table 3) were similar or both groups, but the EVLWI and PVPI were higher in the EVLWI-oriented group.

There were also no significant intragroup changes in MAP, HR, CI, or CVP during the 48 h of de-escalation therapy. The SVV and PPV increased significantly at the end of the de-escalation therapy in both groups (*p* < 0.025). The GEDVI did not change during whole de-escalation period in both groups. The EVLWI values at baseline exceeded 10 mL/kg in 63% of patients. We observed higher values of EVLWI and PVPI in the EVLWI-oriented group during the whole study period. During dehydration therapy, both EVLWI and PVPI decreased significantly only in the EVLWI-oriented group (*p <* 0.02) and remained unchanged in the group guided by the GEDVI. In parallel, we observed the attenuation of metabolic acidosis in both groups (*p <* 0.001). The lactate concentration, as well as albumin and CRP plasma levels, did not change significantly during the 48 h. We observed a transient decrease in potassium at 24 h in the EVLWI-oriented group (*p* = 0.02), whereas sodium rose in both groups during the de-escalation therapy (*p <* 0.05). The plasma creatinine concentration decreased significantly during the EVLWI-guided de-escalation therapy (*p* = 0.03). At 24 and 48 h, urea was higher in the GEDVI group (*p* < 0.02).

After the de-escalation fluid management, PaO_2_/FiO_2_ increased by 30% in the EVLWI group and by 15% in the GEDVI group (*p* < 0.02; Table 3). In the EVLWI-oriented group, we also observed a transient increase in tidal volume at 24 h and in minute volume of ventilation at 48 h, whereas in the GEDVI group, minute ventilation decreased at 24 h (*p <* 0.05 compared with the baseline). There were no intergroup differences in PaCO_2_, minute ventilation, or airway pressures.

We registered higher baseline EVLWI and lower baseline PaO_2_/FiO_2_ ratios in patients with direct ARDS in comparison with ARDS of extrapulmonary origin (Figure 1 and Figure 2). In patients with pulmonary ARDS, we observed significant decrease in EVLWI (*p* = 0.006) and improvement of the PaO_2_/FiO_2_ ratio (*p < 0*.001) during active fluid removal, regardless of the goal-oriented group.

We found a weak correlation between a positive cumulative fluid balance at baseline and a negative outcome at Day 28 (*rho* = 0.3, *p* = 0.038). The decrease in EVLWI during de-escalation therapy was associated with the improvement of the PaO_2_/FiO_2_ ratio (*rho* = 0.3, *p* = 0.03). In addition, the reduced EVLWI during the 48 h of study was associated with a parallel decrease in GEDVI (*rho* = 0.5, *p <* 0.001). During the ROC analysis, a decrease exceeding 2 mL/kg in EVLWI during the 48 h of the de-escalation therapy predicted a positive outcome at Day 28 with an AUC of 0.67, sensitivity of 57%, and specificity of 75% (*p* = 0.02; Figure 3).

## 3. Discussion

Our study has shown that the de-escalation therapy of sepsis and ARDS after initial fluid resuscitation did not compromise hemodynamics or metabolic status; moreover, it can improve arterial oxygenation and organ function.

One of the important parts of our study was the evaluation of the safety of de-escalation therapy. In most investigations, active fluid removal was started only in hemodynamically stable patients [14]; thus, the exclusion criterion in our study was refractory shock with norepinephrine doses exceeding 0.4 mcg/kg/min. We did not observe any worsening of hemodynamic parameters during continuous infusion of furosemide or CVVH; the MAP, CI, and HR remained unchanged in both groups during the 48 h. Furthermore, the number of patients receiving continuous infusion of norepinephrine decreased by twofold by the end of the study in both groups. The attenuation of shock, in parallel with de-escalation therapy, can be explained by a reduced fluid balance and the removal of inflammatory mediators during RRT that can counteract sepsis-induced vasodilatation and myocardial dysfunction [2].

Well-known complications of furosemide infusion are metabolic alkalosis and electrolyte disturbances [15,16]. In our study, active fluid removal led to the attenuation of metabolic acidosis with a normalization of base excess values. Despite a slight increase, sodium concentrations also stayed within the normal range in the majority of the patients, and the decrease in potassium in the EVLWI group was transient. Nevertheless, the changes in electrolyte levels during diuretic therapy required close monitoring and correction, if necessary. The results of a recent pilot study demonstrate that the maximum “safe” dosage of furosemide over 72 h is 200 mg [17]. In our study, the average dosage of furosemide within 48 h was 120 mg, without intergroup differences. Thus, limiting the dose of furosemide reduces the risk of hypernatremia, hypokalemia, and metabolic alkalosis during active fluid removal.

The important risk factor for the development of AKI and low urine output is hypovolemia. However, many cases of AKI are considered volume unresponsive, in particular during systemic inflammation and sepsis [17]. In these situations, injudicious use of fluids carries the additional risk of contributing to the development or worsening of AKI by fluid overload [6]. In our study, we found that active fluid removal in the EVLWI-oriented group led to a significant decrease in creatinine concentration that was accompanied by a reduction in urea compared with the GEDVI group. This is consistent with several publications showing that the use of diuretics may lead to a more rapid resolution of AKI [18]. Thus, Grams et al. reported that in patients with AKI, the cumulative diuretic dose was independently associated with lower mortality [18]. However, in another study, Martin et al. found no difference in serum creatinine changes during the use of diuretics compared with a placebo [19].

Our study has revealed that de-escalation fluid management was accompanied by a decrease in the SOFA scores of both groups. Mainly, it was achieved due to the attenuation of shock, ARDS, and acute kidney injury. Similar findings were obtained by other authors using a de-escalation strategy over a 5-day study period [19]. Such an improvement in organ function can be explained both by the resolution of tissue edema and the optimization of oxygen transport during de-escalation therapy [20].

For the moment, it is still unclear which strategy is better to follow after initial fluid resuscitation: restrictive fluid therapy or active fluid removal. Indeed, more restrictive fluid management, together with earlier administration of vasopressors, if needed, may reduce the volume of fluid therapy [21]. However, it is unlikely that, by using this strategy, fluid overload can be entirely avoided. Fluid intake in an ICU patient is the result of different sources, many of which are obligatory for therapy, such as drug diluents and nutrition. A recent study has shown that this “supportive fluid” accounts for as much as 33% of whole ICU fluid intake compared to 7% for resuscitation fluids [21,22]. Thereby, goal-directed fluid removal could be a method of choice for many ICU patients, especially with ARDS and fluid overload.

Another important issue is the decision about when to start de-escalation fluid management. There is a wide spectrum of clinical criteria for such a decision, including a decrease inPaO_2_/FiO_2_, a high cumulative fluid balance, and an increased GEDVI or EVLWI [6,23,24,25]. In our study, 63% of patients already had signs of lung edema at the beginning of the de-escalation phase. The EVLWI-oriented de-escalation therapy led to the resolution of pulmonary edema and the improvement of oxygenation more efficiently than the GEDVI-guided treatment algorithm. These effects were also accompanied by a reduced PVPI, demonstrating a decreased capillary leak, and were more prominent in patients with direct ARDS. In numerous studies, an EVLWI value exceeding 10 mL/kg, in combination with other factors, was considered an important sign of fluid overload [23,24,25]. Moreover, several experts have proposed an increased EVLWI as an additional criterion for the definition of ARDS [26]. Our findings demonstrate the prognostic value of a decrease in EVLWI exceeding 2 mL/kg during 48 h of de-escalation therapy for a positive outcome. These results are consistent with other authors, showing that the decrease in EVLWI during the first 48 h of treatment was associated with a 28-day survival of ARDS [27]. Moreover, the absolute values of EVLWI with a cut-off point of 12 mL/kg in two days after initial resuscitation were independent predictors of mortality due to sepsis [27].

Dynamic indices of fluid responsiveness, such as PPV or SVV, have better diagnostic value as compared to static variables in fully sedated and mechanically ventilated patients, although both PPV and SVV have a number of limitations, including vasopressor support, low tidal volume, and spontaneous breathing [28,29]. In our study, SVV and PPV increased significantly during de-escalation therapy, although GEDVI remained at the same level, and the number of patients receiving norepinephrine decreased significantly. This can be explained by the restoration of spontaneous breathing activity at the end of our study.

It has been shown that GEDVI is one of the most accurate indicators of preload in patients with sepsis and ARDS, and that GEDVI-oriented fluid therapy prevents the progress of lung edema [30]. In our study, GEDVI-oriented fluid removal was also accompanied by increased PaO_2_/FiO_2_ but, in contrast to the EVLWI-guided group, did not significantly influence the course of pulmonary edema and acute kidney injury. A possible explanation could be the fact that the main mechanism of tissue edema in sepsis-induced ARDS includes increased vascular permeability, which can be observed at normal or even reduced values of GEDVI [6]. Thus, for patients with sepsis, EVLWI can represent a more appropriate goal for dehydration therapy.

We did not observe any intergroup difference in the 48-h fluid balance. The possible explanation of this finding is that we observed a similar pattern of changes in GEDVI and EVLWI after the fluid therapy conducted before the enrollment of 60% of our patients. This circumstance resulted in similar fluid management strategies for both groups.

### 3.1. Pulmonary and Extrapulmonary ARDS

In our study, the patients with direct ARDS had more prominent lung edema and worse arterial oxygenation in comparison with extrapulmonary ARDS. Many researchers acknowledge that pulmonary and extrapulmonary ARDS are not identical with regard to their morphofunctional aspects and responses to PEEP, the recruitment maneuver, prone position, and other adjunctive therapies [31,32,33,34]. Responses to fluid therapy may also differ according to the origin of ARDS. In 2001, Gattinoni and Pelosi, using CT scans, showed that extrapulmonary ARDS manifests with interstitial edema and increased vascular permeability, whereas in pulmonary ARDS, the main reason for hypoxemia is direct damage to the alveolar epithelium [35]. However, according to our data, EVLWI and PVPI were significantly higher in patients with pulmonary ARDS; moreover, the efficacy of active fluid removal was higher in direct ARDS in both groups, independently from the hemodynamic target for de-escalation. These results are consistent with the findings of Morisawa et al. [36]. The direct cause of ARDS, in combination with systemic inflammatory response, may influence pulmonary permeability to a greater extent than in indirect ARDS. Thus, animal studies have suggested that the levels of inflammatory cytokines (interleukins-6, 8, and 10) found in bronchoalveolar lavage fluid were significantly elevated in pulmonary ARDS models compared to extrapulmonary ARDS, whereas no differences were observed in the number of infiltrating neutrophils [37]. These findings suggest that the severity of pulmonary inflammation in patients with direct ARDS may be higher. This assumption is confirmed by a recent study by Coppola et al., which shows, in a manner consistent with our results, that pulmonary ARDS is accompanied by more severe hypoxemia [38].

### 3.2. Fluid Balance

Several studies shown a close relationship between the cumulative balance and the survival rate of patients with sepsis [4,5,6,9,39]. Fluid overload can lead to tissue edema, progression of organ dysfunction, and deterioration due to shock [6]. In our study, we also observed a correlation between a positive fluid balance at baseline and mortality at Day 28. In contrast, the achievement of a negative fluid balance on the third day from the onset of critical illness was associated with an improved clinical outcome [11]. Thus, increased cumulation of fluids may reflect a greater severity of illness and an actual need for a decision either to withhold infusion or to administer diuretics or RRT [40]. It is important to mention that there area lot of confounding factors influencing survival of sepsis, such as effective antibiotic therapy and adequate source control [41]. This can explain the relatively weak correlation between the fluid balance and the outcomes in our study.

### 3.3. Limitations

The study was performed on a small population, which increases the potential bias. Chronic kidney disease was one of the exclusion criteria, which can potentially limit the generalizability of our results. Another important limitation of our study is the heterogeneity of the patients, including the initially lower baseline levels of EVLWI in the GEDVI-oriented de-escalation group. As we did not observe any intergroup difference in the 48 h fluid balance or in the requirement for CVVH, the more prominent pulmonary edema in the EVLWI-oriented group could be an important confounding factor, explaining the decrement of EVLWI and creatinine concentrations during active fluid removal.

## 4. Materials and Methods

The study was performed in a 1000-bed university hospital (City Hospital #1 of Arkhangelsk, Russia). The study protocol and the informed consent form were approved by the Ethics Committee of the Northern State Medical University (Arkhangelsk, Russia). Written informed consent was obtained from the patient or, when unconscious, from the patient’s next of kin. Sixty-five adult patients with sepsis and ARDS were screened for enrollment into a prospective randomized study from 2014 to 2019. Sepsis and ARDS were diagnosed according to the third international definition of sepsis and septic shock and the Berlin definition of ARDS, respectively [41,42,43]. The inclusion criteria were the presence of sepsis and ARDS as described above, mechanical ventilation before the study for at least 24 h, and the age of the patient >18 years. The exclusion criteria were the requirement for continuous infusion of norepinephrine in a dose exceeding 0.4 mcg/kg/min to maintain mean arterial pressure (MAP) within 65–75 mm Hg, morbid obesity with a BMI > 40 kg/m^2^, severe brain injury, chronic kidney diseases, pregnancy, and known irreversible underlying conditions such as end-stage neoplasms. The study flow chart is presented in Figure 4. Five patients were excluded during the screening phase.

At the study baseline, the enrolled patients were randomly assigned to a strategy of fluid therapy using either extravascular lung water index (EVLWI)-oriented (*n* = 30) or global end-diastolic volume index (GEDVI)-oriented de-escalation therapy (*n* = 30) using an envelope method. Patients received de-escalation fluid management using either diuretics or ultrafiltration during continuous RRT in cases of GEDVI > 650 mL/m^2^ or EVLWI > 10 mL/kg, depending on the randomization group. The primary goal of de-escalation was to obtain a cumulative fluid balance of 0 to −3000 mL after 48 h from the study baseline. In cases of GEDVI < 650 mL/m^2^ or EVLWI < 10 mL/kg, the target fluid balance was in the range of 0 to +3000 mL (Figure 1). Only patients were blind to group allocation. Care providers and investigators could not be blind due to the presence of the hemodynamic monitor.

In all patients, we catheterized the internal jugular or the subclavian vein with a triple-lumen central venous catheter (Certofix, B|Braun, Melsungen, Germany) and the femoral artery with a thermistor-tipped arterial catheter (5F, PV2015L20, Pulsion Medical Systems, Munich, Germany). The arterial blood pressure was recorded from a side port of the catheter. Hemodynamic monitoring was carried out using the method of transpulmonary thermodilution (PiCCO_2_ monitor, Pulsion Medical Systems, Munich, Germany) by a triplicate 15 mL bolus of cold (<8 °C) 0.9% saline solution. Mechanical ventilation (Puritan Bennett 840 and 980, Medtronic, USA) was performed according to the ARDS Network protocol and the Surviving Sepsis Campaign, using pressure-controlled ventilation with a tidal volume of 6–7 mL/kg of predicted body weight, with a Ppeak not exceeding 35 cm H_2_O, positive end-expiratory pressure (PEEP), and a fraction of inspiratory oxygen (FiO_2_) levels adjusted to maintain SpO_2_ within a 92–97% range [1,43]. We started weaning using synchronized intermittent pressure-controlled ventilation with pressure support on patients with PaO_2_/FiO_2_>200 mm Hg. The weaning protocol included gradual reduction of FiO_2_, a mandatory respiratory rate, and an inspiratory pressure with a subsequent spontaneous breathing trial followed by tracheal extubation.

The target fluid balance was achieved by continuous infusion of furosemide with an initial dose of 0.07 mcg/kg/h and a minimal duration of 12 h. In case of failure to reach a negative fluid balance by means of diuretics in patients with AKI and septic shock, we started controlled ultrafiltration using continuous veno-venous hemofiltration (CVVH, multiFiltrate, Fresenius Medical Care, Bad Homburg, Germany). For fluid replacement, if necessary, we administered balanced crystalloid solutions (Sterofundin Iso/G5, B|Braun, Melsungen, Germany) with an initial infusion rate of 6–7 mL/kg/h.

Heart rate (HR), MAP, cardiac index (CI), GEDVI, EVLWI, pulmonary vascular permeability index (PVPI), central venous pressure (CVP), systemic vascular resistance index (SVRI), pulse pressure variation (PPV), and stroke volume variation (SVV) were assessed using transpulmonary thermodilution and arterial pulse contour analysis. During the study, we also assessed blood gases (ABL Flex 800 Radiometer, Copenhagen, Denmark) and biochemical parameters (Random Access A-25, BioSystems, Barcelona, Spain). All measurements were performed at baseline, at 24, and at 48 h of the goal-directed de-escalation.

The clinical characteristics assessed during the study included fluid balance, doses of vasopressors, duration of mechanical ventilation, and length of ICU and hospital stays. We evaluated the severity of organ dysfunction at the study entry and at 48 h according to the Sequential Organ Failure Score (SOFA) [14]. The 28-day mortality was obtained from the patient records.

The results of the study were partially published in Russian [44]. The current publication includes the additional analyses of the hemodynamic, biochemical, and clinical data, as well as details of the dehydration therapy. The primary study endpoint was to assess whether de-escalation therapy improves organ function. According to our preliminary data analysis, it was assumed that the EVLWI-oriented strategy is associated with a decrease of 20% in the SOFA score over 48 h. Assuming a two-sided α level of 0.05 and a study power of 0.8, we calculated that a sample size of 25 patients would be required in each group. Finally, 60 patients were enrolled to allow for dropouts. The secondary endpoint was to evaluate the influence of de-escalation algorithms on mortality.

### Statistical Analysis

For data collection and analysis, we used SPSS software (version 17.0; SPSS Inc., Chicago, IL, USA). All the variables were expressed as a median (25th–75th percentiles). The groups were compared using a Mann–Whitney test. The intragroup comparisons were performed using the Wilcoxon test. Nominal data were compared using a χ^2^-test or a Fisher exact test and expressed as patient numbers or %. The correlation analysis was performed using Spearman’s *rho*. To evaluate the prognostic value of the decrease in EVLWI during the 48 h of de-escalation therapy for 28-day survival, we performed a receiver operating characteristic (ROC) curve analysis and calculated the area under the ROC curve (AUC). The optimal cut-off point for EVLWI was determined by the maximum value of the Youden Index (maximizing sensitivity and specificity). A *p* value < 0.05 was regarded as statistically significant.

## 5. Conclusions

In sepsis-induced ARDS, both fluid management strategies based either on GEDVI or EVLWI improve arterial oxygenation and attenuate organ dysfunction. Active fluid removal is more efficient in patients with pulmonary ARDS, regardless of the choice of volumetric targets for de-escalation. Further studies of adequately controlled goal-directed protocols of fluid therapy for patients with sepsis and ARDS are warranted.

## Figures and Tables

**Figure 1 ijms-24-08768-f001:**
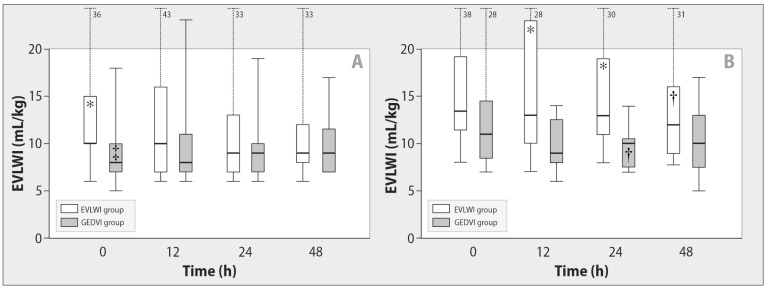
Changes in extravascular lung water index during de-escalation therapy in patients with sepsis-induced indirect (**A**) and direct (**B**) ARDS. Data are expressed as a median (25th–75th percentiles). EVLWI—extravascular lung water index, GEDVI—global end-diastolic volume index. * *p* ≤ 0.05 for comparison with the GEDVI group; ^†^
*p <* 0.03 for intragroup comparison with baseline; ^‡^
*p <* 0.05 for comparison with direct ARDS at each stage.

**Figure 2 ijms-24-08768-f002:**
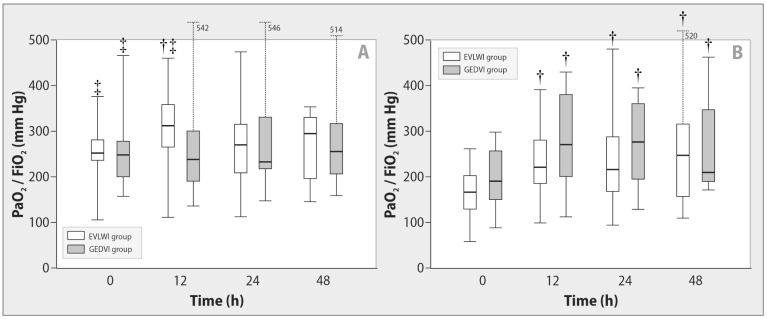
Changes in the PaO_2_/FiO_2_ ratio during de-escalation therapy in patients with sepsis-induced indirect (**A**) and direct (**B**) ARDS. Data are expressed as a median (25th–75th percentiles). EVLWI—extravascular lung water index, GEDVI—global end-diastolic volume index.^†^
*p <* 0.02 for intragroup comparison with baseline; ^‡^
*p <* 0.03 for comparison with direct ARDS at each stage.

**Figure 3 ijms-24-08768-f003:**
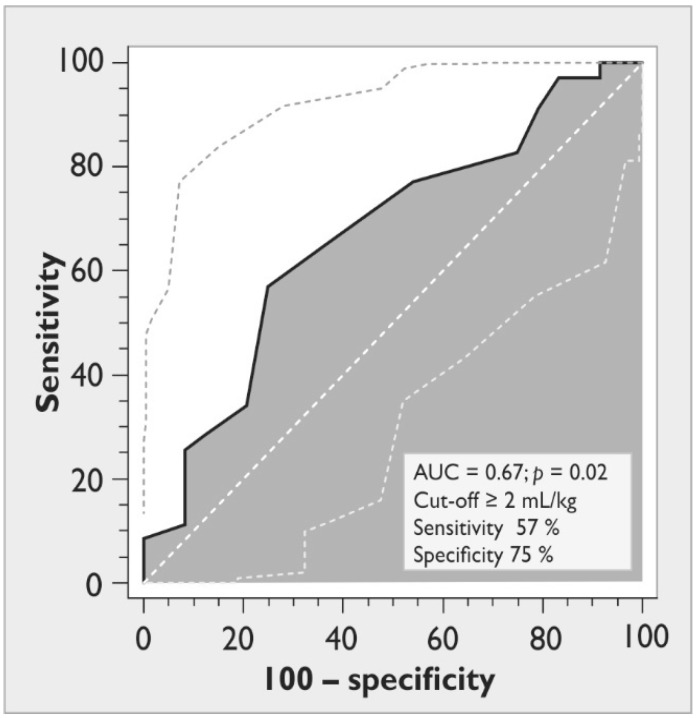
The ROC curve for the decrease in extravascular lung water index during 48 h of de-escalation therapy as a predictor of 28-day survival. Dashed straight line represents random classifier; solid curved line represents received operating characteristic curve; dashed curved lines represent 95% confidence interval; gray zone represents area under curve.

**Figure 4 ijms-24-08768-f004:**
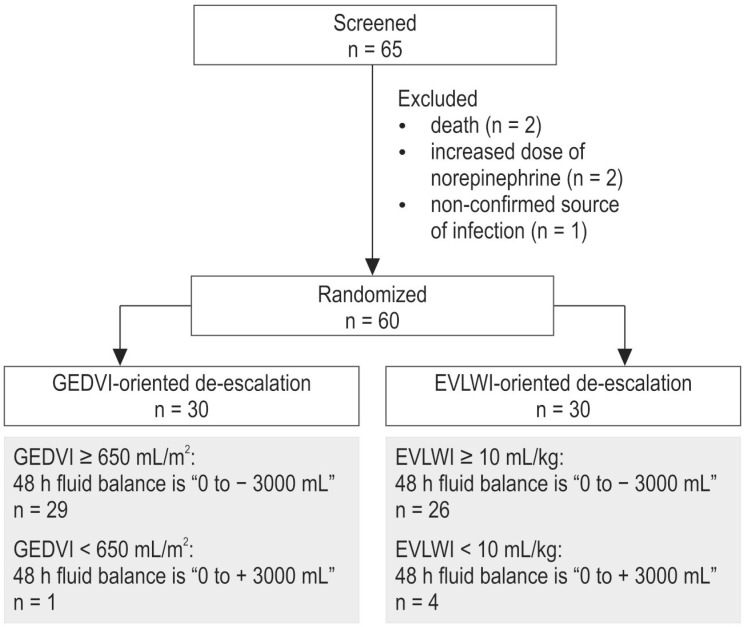
Study flow chart. EVLWI—extravascular lung water index, GEDVI—global end-diastolic volume index.

**Table 1 ijms-24-08768-t001:** Baseline characteristics of the patients.

Characteristics	Group		*p*
	EVLWI (*n* = 30)	GEDVI (*n* = 30)	
Age, years	54 (24–79)	58 (30–86)	0.4
Sex, male/female	16/14	20/10	0.3
Origin of sepsis, *n* (%):			
pneumonia	19 (63.3)	13 (43.3)	0.2
necrotizing pancreatitis	5 (16.7)	7 (23.3)	0.7
peritonitis	2 (6.7)	9 (30.0)	0.04
mediastinitis	2 (6.7)	–	0.5
pyelonephritis	1 (3.3)	1(3.3)	1.0
soft tissue infection	1 (3.3)	–	1.0
Direct ARDS, *n* (%)	19 (63.3)	13 (43.3)	0.2
Indirect ARDS, *n* (%)	11 (36.7)	17 (56.7)	0.2
ARDS severity, *n* (%):			
mild	14 (46.6)	17 (56.6)	0.6
moderate	14 (46.6)	12 (40.6)	0.8
severe	2 (6.8)	1 (2.8)	1.0
Components of organ failure, *n* (%):			
septic shock	10 (33)	11 (37)	1.0
DIC	17 (57)	19 (63)	0.8
acute liver failure	8 (27)	11 (37)	0.6
acute renal failure	22 (73)	18 (60)	0.4
SOFA score, baseline, pt.	8 (3–14)	8 (4–14)	0.5
SOFA score, 48 h, pt.	6 (1–14) ^†^	7 (1–16) ^†^	0.1
Incidence of norepinephrine administration, baseline, n (%)	19 (63)	21 (70)	0.6
Incidence of norepinephrine administration, 48 h, *n* (%)	9 (30) ^†^	12 (40) ^†^	0.4
Average dose of furosemide within 48 h, mg	120 (120–320)	120 (120–180)	0.8
Incidence of CVVH, *n* (%)	12 (40)	11 (37)	0.6
Fluid balance, baseline, mL	690(−212 to +1512)	430(−75 to +1725)	0.9
Fluid balance, 48 h, mL	−2210(−3020 to −1573) ^†^	−2298(−2982 to −1616) ^†^	0.9
Time in ICU before randomization, days	2 (1–4)	3 (1–4)	0.3
Duration of mechanical ventilation, days	9 (4–16)	7 (4–17)	0.7
Duration of ICU stay, days	16 (9–22)	12 (8–20)	0.4
Duration of hospitalization, days	24 (16–46)	24 (15–32)	0.4
28-day mortality, *n* (%)	10 (33.3)	14 (46.7)	0.3

EVLWI—extravascular lung water index, GEDVI—global end-diastolic volume index, ARDS—acute respiratory distress syndrome, DIC—disseminated intravascular coagulation, SOFA—Sequential Organ Failure Assessment, CVVH—continuous veno-venous hemofiltration. Data are presented as a median and 25th–75th percentiles or numbers (%). ^†^
*p <* 0.05 for intragroup comparison with baseline.

**Table 2 ijms-24-08768-t002:** Clinical and laboratory parameters during de-escalation therapy.

Characteristics	Group	Stages		
		Baseline	24 h	48 h
MAP, mm Hg	EVLWI	82 (71–94)	77 (66–89)	80 (73–90)
	GEDVI	80 (73–85)	80 (72–84)	83 (76–96)
HR, bpm	EVLWI	100 (80–111)	89 (79–104)	94 (78–104)
	GEDVI	92 (79–110)	89 (78–109)	93 (84–107)
CI, L/min/m^2^	EVLWI	3.6 (3.0–4.2)	3.4 (2.9–4.4)	3.7 (3.1–4.4)
	GEDVI	3.7 (3.1–4.2)	3.8 (3.3–4.3)	3.9 (3.5–4.5)
CVP, mm Hg	EVLWI	10 (8–12)	7 (5–10)	8 (5–10)
	GEDVI	9 (7–13)	9 (7–11)	10 (8–12)
GEDVI, mL/m^2^	EVLWI	816 (642–951)	768 (636–956)	751 (602–1005)
	GEDVI	776 (701–902)	763 (708–903)	778 (702–908)
EVLWI, mL/kg	EVLWI	13 (10–18)	12 (9–15)	11 (8–16) ^†^
	GEDVI	9 (7–11) *	9 (7–12) *	9 (7–12)*
PVPI	EVLWI	2.7 (2.2–3.2)	2.6 (2.1–3.2)	2.5 (2.0–3.0) ^†^
	GEDVI	1.8 (1.5–2.1) *	1.6 (1.4–1.8) *	1.8 (1.6–2.0) *
PPV, %	EVLWI	10 (6–17)	16 (9–19)	15 (8–18)
	GEDVI	8 (5–13)	10 (8–15)	13 (7–16) ^†^
SVV, %	EVLWI	10 (6–20)	16 (9–19)	15 (11–20) ^†^
	GEDVI	9 (6–13)	11 (8-17)	13 (9–19)
BE, mmol/L	EVLWI	−3.1 (−7.3 to −0.8)	−0.8 (−3.5 to +2.8) ^†^	−0.8 (−2.1 to +1.5) ^†^
	GEDVI	−4.1 (−6.3 to −0.7)	−1.3 (−3.4 to +0.7) ^†^	−0.4 (−2.8 to +1.6) ^†^
Lactate, mmol/L	EVLWI	1.7 (1.1–3.0)	2.0 (1.5–2.8)	1.7 (1.2–2.6)
	GEDVI	1.8 (1.2–2.9)	2.0 (1.6–2.3)	1.9 (1.4–3.0)
Albumin, g/L	EVLWI	28 (25–33)	28 (23–31)	29 (23–32)
	GEDVI	26 (22–29)	27 (23–29)	24 (18–32)
CRP, mg/L	EVLWI	203 (114–370)	218 (116–378)	230 (124–361)
	GEDVI	194 (164–410)	191 (153–381)	198 (120–398)
Potassium, mmol/L	EVLWI	4.0 (3.6–4.4)	3.5 (3.3–4.0) ^†,^*	4.0 (3.6–4.4)
	GEDVI	4.2 (3.8–4.5)	4.3 (3.7–4.6)	4.2 (3.8–4.6)
Sodium, mmol/L	EVLWI	141 (137–145)	144 (139–146) ^†^	143 (139–151) ^†^
	GEDVI	141 (137–147)	143 (138–150) ^†^	144 (138–149)
Creatinine, mcmol/L	EVLWI	126 (98–168)	119 (98–158)	107 (93–142) ^†^
	GEDVI	123 (79–179)	136 (88–175)	133 (85–161)
Urea, mmol/L	EVLWI	9 (8–12)	10 (8–12)	8 (6–12)
	GEDVI	12 (8–19)	13 (9–19)*	13 (9–17)*

MAP—mean arterial pressure, HR—heart rate, CI—cardiac index, CVP—central venous pressure, GEDVI—global end-diastolic volume index, EVLWI—extravascular lung water index, PVPI—pulmonary vascular permeability index, PPV—pulse pressure variation, SVV—stroke volume variation, BE—base excess, CRP—C-reactive protein. Data are presented as a median (25th–75th percentiles). ^†^*p <* 0.05 for intragroup comparison with baseline; * *p <* 0.05 compared with the EVLWI-oriented group.

**Table 3 ijms-24-08768-t003:** Respiratory parameters during de-escalation therapy.

Characteristics	Group	Stages		
		Baseline	24 h	48 h
PaO_2_/FiO_2_,mm Hg	EVLWI	195 (133–253)	241 (168–310)^†^	254 (159–319) ^†^
	GEDVI	217 (185–272)	258(215–341)^†^	248 (194–330) ^†^
PaCO_2_, mm Hg	EVLWI	37 (32–46)	36 (31–39)	37 (33–42)
	GEDVI	36 (31–42)	36 (32–40)	38 (33–40)
Tidal volume, mL	EVLWI	495 (443–607)	620 (555–664) ^†^	582 (509–634)
	GEDVI	541 (478–648)	507 (436–617) *	506 (444–637)
Minute volume of ventilation, L	EVLWI	9.9 (8.1–12.2)	10.7 (8.7–12.5)	12.1 (9.8–13.9) ^†^
	GEDVI	10.4 (8.2–13.7)	9.6 (7.9–11.2) ^†^	9.9 (8.6–12.4)
PEEP, cm H_2_O	EVLWI	8 (8–10)	8 (7–11)	8 (6–10)
	GEDVI	8 (7–10)	8 (6–10)	8 (6–10)
Mean airway pressure, cm H_2_O	EVLWI	13 (12–15)	13 (11–16)	13 (10–16)
	GEDVI	14 (12–16)	14 (11–15)	14 (11–16)

PEEP—positive end-expiratory pressure.Data are presented as a median (25th–75th percentiles). ^†^
*p <* 0.05 for intragroup comparison with baseline; * *p <* 0.05 compared with the EVLWI-oriented group.

## Data Availability

The data is unavailable for public access due to hospital privacy restriction and can be provided on the individual basis via e-mail.

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
