# Peer review of "Optimizing Fluid Management Guided by Volumetric Parameters in Patients with Sepsis and ARDS"

_ijms, 2023, doi:10.3390/ijms24108768_

Round 1
Reviewer 1 Report
I read with great interest the manuscript entitled "Optimizing fluid management guided by volumetric parameters in patients with sepsis and ARDS". The authors note that achieving an optimal fluid status in critically ill patients is a challenge in current critical care practice. The investigators conducted a study to compare two different de-escalation strategies based upon either extravascular lung water or global end-diastolic volume in patients with sepsis and ARDS. The study was conducted in a 1000 bed hospital in Russia. Patients with sepsis and ARDS were screened and randomized to the 2 separate de-escalation strategies.
– The study population is small with only 60 patients included in the study, which increases the potential for bias and also increases the likelihood that findings are due to chance.
– The study was conducted over a period of 5 years which introduces the possibility of bias due to evolution of the care of patients with sepsis and ARDS over time due to new evidence and technology.
– Patients with morbid obesity and chronic kidney disease were excluded from the study which limits generalizability of the results.
– It is unclear and not clearly specified within the manuscript when de-escalation strategies were initiated and how it was determined that a patient was an appropriate to begin de-escalation. This is often the biggest question in the ICU: When is it time to begin de-resuscitation or initiate a de-escalation strategy? This was not standardized in the study and could significantly impact the findings.
– Baseline characteristics of both groups were similar but do show some heterogeneity and could impact the observed findings. For example more patients in the EVLWI group had pneumonia and significantly more patients in the GEDVI group had peritonitis. Patients in the GEDVI group were also in the ICU for a longer period of time prior to randomization which could have had an impact on the findings.
–Given both de-escalation strategies achieved a similar fluid balance at 48 hours and received similar doses of furosemide, the findings of decreased sofa scores and decreased number of patients requiring norepinephrine infusion are unlikely due to the type of de-escalation strategy used.
–The authors concluded that one de-escalation strategy attenuates pulmonary edema and acute kidney injury more efficiently than the other however both strategies of de-escalation achieved a similar fluid balance at 48 hours. How can we attribute one strategy is better than the other when both achieved a similar fluid balance?
–Based on the previous 2 points, further clarification is needed on how the authors got to their conclusion that one strategy is better than the other when both strategies achieve a similar fluid balance of 40 hours and both groups also received a similar amount of furosemide. At this point I do not believe the conclusions are appropriately supported by the results.
Author Response
Thank you so much for your interest to our study and valuable comments.
- The study population is small with only 60 patients included in the study, which increases the potential for bias and also increases the likelihood that findings are due to chance.
Thank you for your comments; we included this information in the chapter of limitations at the end of Discussion.
- The study was conducted over a period of 5 years which introduces the possibility of bias due to evolution of the care of patients with sepsis and ARDS over time due to new evidence and technology.
Although we enrolled patients during 5 years, we did not observe any major changes in local clinical practice regarding mechanical ventilation, phase approach to fluid therapy and de-escalation strategy for antibiotic therapy. As we have indicated in Methods, during recruitment period the therapy of patients was performed according to Surviving Sepsis Campaign recommendations 2016.
- Patients with morbid obesity and chronic kidney disease were excluded from the study which limits generalizability of the results.
Thank you for your comments; we included this information in the limitation chapter. As one from the tasks of our study was to assess the role of furosemide in dehydration strategy, it was important for us to exclude patients with chronic kidney disease.
- It is unclear and not clearly specified within the manuscript when de-escalation strategies were initiated and how it was determined that a patient was an appropriate to begin de-escalation. This is often the biggest question in the ICU: When is it time to begin de-resuscitation or initiate a de-escalation strategy? This was not standardized in the study and could significantly impact the findings.
Thank you for this valuable comment. We absolutely agree with you that it is the biggest question in ICU. As mentioned in inclusion criteria, we started our therapy after at least 24- hr stay in ICU (time for initial resuscitation therapy) and when the dose of norepinephrine was ≤ 0.4 mcg/kg/min. It was important for us to demonstrate that even with such simplified criteria the de-escalation therapy was safe and did not lead to deterioration of shock.
- Baseline characteristics of both groups were similar but do show some heterogeneity and could impact the observed findings. For example more patients in the EVLWI group had pneumonia and significantly more patients in the GEDVI group had peritonitis. Patients in the GEDVI group were also in the ICU for a longer period of time prior to randomization which could have had an impact on the findings.
We agree with you, thus we performed the post hoc analysis to assess the effects of de-escalation therapy in patients with direct (pneumonia) and indirect (peritonitis) ARDS. We have also indicated the heterogeneity of our patients as one from study limitations in the Discussion.
- –Given both de-escalation strategies achieved a similar fluid balance at 48 hours and received similar doses of furosemide, the findings of decreased sofa scores and decreased number of patients requiring norepinephrine infusion are unlikely due to the type of de-escalation strategy used.
Thank you for your comments. The similar fluid balance during study period in both groups was our major concern as well. The fluid status of our patients at study inclusion was different. The initial fluid therapy increased values of both GEDVI and EVLWI in 55% of patients. In 5% of patients we still observed hypovolemia, and only in 40% of patients we found the discordant changes in GEDVI and EVLWI (12% - EVLWI exceeding 10 mL/kg PBW with normal values of GEDVI; 28% - increased GEDVI only with EVLWI < 10 mL/kg PBW). We have added this information into Results and Discussion. Thus, the goal-directed strategy of fluid management could influence the fluid status and organ function of our patients in personalized directions despite similar mean fluid balance in both groups.
- –The authors concluded that one de-escalation strategy attenuates pulmonary edema and acute kidney injury more efficiently than the other however both strategies of de-escalation achieved a similar fluid balance at 48 hours. How can we attribute one strategy is better than the other when both achieved a similar fluid balance?
We have answered this question partially responding to Comment 6. We agree that can’t attribute one strategy is clearly better than the other, thus we changed the conclusion of our study to “In sepsis-induced ARDS, both fluid management strategies based either on GEDVI or EVLWI improve arterial oxygenation and attenuate organ dysfunction”.
- –Based on the previous 2 points, further clarification is needed on how the authors got to their conclusion that one strategy is better than the other when both strategies achieve a similar fluid balance of 40 hours and both groups also received a similar amount of furosemide. At this point I do not believe the conclusions are appropriately supported by the results.
Thank you, as we have indicated earlier, we have followed your advice and changed the conclusion.

Reviewer 2 Report
It has been a pleasure to review this manuscript. Presented information appear clear, precise and well written. However, conclusions are not supported by results.
- The two target fluid balances (from 0 to -3000 and from 0 to +3000 ml) appear large and almost overlapping. Is it possible that this could affect the results?
- Could you please specify how EVLWI and GEDVI are calculated?
- Line 169: table 2 and 3 do not show baseline characteristics.
- Baseline characteristics: EVLWI and GEDVI parameters and PaO2/FiO2 and tidal volume are not presented in table 1. Could authors present them? From table 2, it seems that these two parameters are different at baseline in the two groups: this could have affected the subsequent changes at 24 and 48 hours. In particular, it seems that at baseline EVLWI, GEDVI and PaO2/FiO2 are better at baseline in GEDWI group: could this explain partially their lower change during the study period (48 hours)? In this case, this conclusion may be not supported by data: The EVLWI-oriented de-escalation therapy led to the resolution of pulmonary edema and the improvement of oxygenation more efficiently than the GEDVI-guided treatment algorithm.
o This is partially admitted at the end of the discussion. But this aspect deserves more attention because it significantly alters the significance of results. Conclusions are not supported by results.
Author Response
Thank you so much for your comments and advice.
- The two target fluid balances (from 0 to -3000 and from 0 to +3000 ml) appear large and almost overlapping. Is it possible that this could affect the results?
Thank you for your comment. At study enrolment, we observed different patterns of fluid status. The increased values of both GEDVI and EVLWI were registered in 55% of patients. Five percent of patients had GEDVI < 650 mL/m2 and EVLWI < 10 mL/kg, and only in 40% of patients we found the discordant changes in GEDVI and EVLWI (12% with EVLWI ≥ 10 mL/kg PBW and GEDVI < 650 mL/m2; 28% with GEDVI ≥ 650 mL/m2 and EVLWI < 10 mL/kg PBW). Thus, our patients could require different fluid interventions with different degrees of either fluid load or dehydration depending on their volumetric values. These data are included into revised manuscript.
- Could you please specify how EVLWI and GEDVI are calculated?
The EVLW is a volumetric parameter quantifying pulmonary edema. EVLW is calculated as the difference between ITTV and ITBV, where ITTV (intrathoracic thermal volume) is calculated as Cardiac Output × MTti (mean transit time) of thermoindicator, and ITBV (intrathoracic blood volume) is calculated as 1.25 × GEDV. GEDV is calculated as the difference between ITTV and PTV, where PTV is Cardiac Output × DSt (downslope time) of thermodilution curve.
Calculation of the selected transpulmonary thermodilution-derived hemodynamic variables.
|
Variable |
Calculation |
|
Intrathoracic thermal volume (ITTV) |
CO × MTt |
|
Pulmonary thermal volume (PTV) |
CO × DSt |
|
Global end-diastolic volume (GEDV) |
ITTV – PTV |
|
Intrathoracic blood volume (ITBV) |
1.25 × GEDV |
|
Stroke volume (SV) |
CO / HR |
|
Global ejection fraction (GEF) |
(4 × SV) / GEDV |
|
Cardiac function index (CFI) |
CO / GEDV |
|
Cardiac power index (CPI) |
MAP × CO |
|
Extravascular lung water (EVLW) |
ITTV – ITBV |
|
Pulmonary blood volume (PBV) |
ITBV – GEDV |
|
Pulmonary vascular permeability index (PVPI) |
EVLW / PBV |
CO — cardiac output; MTt — mean transit time; DSt — downslope time; HR — heart rate; ITTV — intrathoracic thermal volume; PTV — pulmonary thermal volume; GEDV — global end-diastolic volume; EVLW — extravascular lung water; PVPI — pulmonary vascular permeability index; PBV — pulmonary blood volume; ITBV — intrathoracic blood volume; MAP — mean arterial pressure.
These calculations are presented in Ref. 27 and 44 of our paper:
- Tagami T, Ong M. Extravascular lung water measurements in acute respiratory distress syndrome: why, how, and when? Curr Opin Crit Care 2018; 24 (3): 209–215.
- Kirov M, Kuzkov V, Saugel B (Eds). Advanced Hemodynamic Monitoring: Basics and New Horizons. Springer, Cham 2021. ISBN 978-3-030-71751-3. ISBN 978-3-030-71752-0 (eBook). https://doi.org/10.1007/978-3-030-71752-0 P. 1-298.
- Line 169: table 2 and 3 do not show baseline characteristics.
Table 2 includes baseline characteristics of hemodynamic and laboratory parameters (column 3). Table 3 includes baseline characteristics of respiratory parameters (column 3).
- Baseline characteristics: EVLWI and GEDVI parameters and PaO2/FiO2 and tidal volume are not presented in table 1. Could authors present them? From table 2, it seems that these two parameters are different at baseline in the two groups: this could have affected the subsequent changes at 24 and 48 hours. In particular, it seems that at baseline EVLWI, GEDVI and PaO2/FiO2 are better at baseline in GEDWI group: could this explain partially their lower change during the study period (48 hours)? In this case, this conclusion may be not supported by data: The EVLWI-oriented de-escalation therapy led to the resolution of pulmonary edema and the improvement of oxygenation more efficiently than the GEDVI-guided treatment algorithm.
Since these baseline characteristics are required in Tables 2 and 3 to compare them with 24 and 48 hrs in the same Tables, we presented them in Tables 2 and 3 only but not in Table 1 to prevent duplicate information. According to statistical analysis, the baseline PaO2/FiO2 ratio, tidal volume and GEDVI did not differ between groups, although EVLWI was significantly lower in GEDVI-group. We agree that the heterogeneity of our patients is an important limitation of our study and included this information into Discussion. We have also changed the conclusion of our study and removed the sentence “The EVLWI-oriented de-escalation therapy led to the resolution of pulmonary edema and the improvement of oxygenation more efficiently than the GEDVI-guided treatment algorithm”.
- This is partially admitted at the end of the discussion. But this aspect deserves more attention because it significantly alters the significance of results. Conclusions are not supported by results.
We have changed the conclusion of our study, as requested.

Round 2
Reviewer 1 Report
Authors have addressed my questions and comments.
Author Response
Thank you so much for your comments and your work
Reviewer 2 Report
We thank the authors for their work. However, we believe that difference between two groups at baseline should be more explicit. In fact, in lines 169-170 it is written: At the baseline, both groups had similar hemodynamic and respiratory parameters 170 (Tables 2 and 3). This is not true for EVLWI which is significantly higher in EVLWI group. This is why, in my opinion, this conclusion is not supported by the results (lines 237-238): The comparison of two treatment algorithms demonstrates that the therapy guided by EVLWI attenuates pulmonary edema and acute kidney injury more efficiently than the GEDVI-oriented algorithm. This is repeated also in the conclusion (lines 369-371): The EVLWI-oriented dehydration protocol using diuretics and/or CVVH is accompanied by a better resolution of lung edema and acute kidney injury in comparison with GEDVI-guided strategy.
Author Response
Thank you so much for Your comments. We have added additional information in lines 170-171 regarding the baseline differences in EVLWI and PVPI. The conclusions mentioned in Your letter are removed.